# Assessment of practices and barriers toward nasogastric tube rehydration for moderate and severe dehydration due to diarrheal disease in under-five children among health centers in Gamo Zone, Ethiopia

Tsegazeab Ayele[1]*, Muluken Birhanu[2], Solomon Kassa[3], Sayih Mehari Degualem[3], Habtamu Wondmagegn[1], Kidus Temesgen[4], Habtamu Esubalew Bezie[4], Tamirayehu Abayneh[5], Maycas Gembe[6], Biniyam Demisse Andarge[3]

1 Department of Anatomy, Arba Minch University, Arba Minch, Ethiopia, 2 Department of Pediatrics, Arba Minch University, Arba Minch, Ethiopia, 3 Department of Nursing, Arba Minch University, Arba Minch, Ethiopia, 4 Department of Public Health, Arba Minch University, Arba Minch, Ethiopia, 5 Department of Nursing, Arba Minch Health Science College, Arba Minch, Ethiopia, 6 Department of Nursing, Mattu University, Mattu, Ethiopia

* tsegazeabayele@gmail.com

## Abstract

### Introduction

Dehydration from diarrheal diseases remains a common cause of morbidity and mortality in developing countries. Nasogastric tube is an easy, efficient, and cost-effective method of management that could be a key to minimizing deaths from diarrheal disease. As such, this study assessed the current practices and barriers toward using nasogastric tube for managing moderate to severe dehydration in under-five children.

### Purpose

To assess the practices and barriers toward the management of moderate to severe dehydration using nasogastric tube in under-five children among health centers in Gamo Zone, South Ethiopia.

### Methods

A qualitative study design was conducted among healthcare professionals at selected health centers in Gamo Zone. Data about the practice pattern of managing moderate to severe dehydration and barriers toward nasogastric tube utilization were obtained via in-depth interview. Data were reviewed using constant comparative analysis to identify emerging themes, and axial coding was performed to make connections between categories to organize themes into causal relationships. Hypotheses and concepts were developed inductively from the data.

**Data availability statement:** All relevant data are within the manuscript.

**Funding:** The author(s) received no specific funding for this work.

**Competing interests:** The authors have declared that no competing interests exist.

## Results

In our study, of thirty participants, 21 (70.0%) correctly diagnosed severe dehydration, while 9 (30.0%) diagnosed moderate dehydration. Among those who diagnosed severe dehydration, 5 (23.8%) recommended intravenous resuscitation, and 16 (76.2%) chose oral rehydration. After being informed of failed attempts, four chose to refer, and one clinician remained fixed on intravenous resuscitation, while the remaining 16 clinicians opted for nasogastric tube. Major challenges mentioned by the participants in managing dehydration were late presentation of the patients and equipment shortage.

## Conclusion

Participants in our study were aware of the significance of nasogastric tube for the treatment of moderate to severe dehydration. However, a gap in clinical skills, a lack of continuous training, high clinician turnover, and institutional policies limiting nasogastric tube utilization for the management of dehydration were major challenges. Therefore, improving clinicians' skills through continuous training and improving management protocol is essential in ensuring safe and effective rehydration and better patient outcomes.

## Introduction

Globally, 38 children out of 1000 live births died before reaching five years old by 2019. Half of the total deaths were from five countries, including Ethiopia, with 178 deaths per 1000 live births. The primary causes of these deaths were infectious diseases, including pneumonia, malaria, and diarrhea [1]. The burden of diarrheal disease persists; it was reported as the second most common cause of death in under-five children globally by 2017 [2]. Particularly, in developing countries, dehydration due to diarrheal disease contributed to high morbidity and mortality in under-five children. In Ethiopia, diarrheal disease accounted for 8.98% of deaths in under-five children making it the third most common cause of mortality in this age group [3]. Additionally, 9.96% of hospital admissions in under-five children were attributed to acute gastritis with dehydration [4].

Even though diarrheal disease is a preventable and treatable health issue, high mortality rates may be attributed to gaps in the implementation of the WHO's management protocols for dehydration [5]. A study has reported that only 9% of clinicians in western Kenya followed the guidelines to treat severe dehydration using a nasogastric tube (NGT). Most clinicians remained fixed on IV resuscitation even after two failed attempts to access a vein [6]. In resource-limited settings where trained personnel and necessary equipment are scarce, this reliance on IV therapy could leave clinicians without any treatment options, risking children's lives as the inability to secure an IV line can delay life-saving interventions.

Despite these challenges, in Ethiopia, experienced clinicians are commonly promoted to urban areas, while less experienced clinicians are appointed to remote rural areas where diarrheal diseases are more prevalent. This discrepancy further complicates IV fluid administration; as less experienced clinicians may struggle with securing an IV line. Additionally, many health centers (HC) are located more than a 30-minute drive from hospitals, delaying referrals; furthermore, some parents may refuse to transport children to distant facilities due to logistical or financial constraints. Referral challenges in Ethiopia are reflected by a minimal number of referred cases and suggestions for improved connections between hospitals and primary healthcare units [7].

On the other hand, studies have shown that NGT is as effective as IV therapy for moderate to severe dehydration [8,9]. In fact, moderately dehydrated children had a better outcome with NGT fluid replacement using 50 ml/kg non-flavored oral rehydration solution (ORS) over three hours than with IV 50 ml/kg normal saline resuscitation within three hours, according to both clinical and laboratory biomarkers [8]. Furthermore, children with moderate to severe dehydration who would typically be managed by IV fluid replacement were managed effectively using NGT [10].

Rehydration using NGT is indicated when children with mild to moderate dehydration are unable to tolerate oral rehydration therapy (ORT) due to persistent vomiting, refuse to drink, or when there is concern about the family's ability to effectively administer ORT [10]. Additionally, NGT is recommended when immediate access to IV resuscitation is not possible, such as when the nearest hospital is more than a 30-minute drive away [9]. Besides, NGT is easier to administer, has a lower failure rate, and is more cost-efficient than IV therapy while maintaining similar efficacy [8]. In contrast, securing an IV line in dehydrated children is challenging and often requires multiple attempts, which delays treatment, causes pain, and increases the risk of parenteral inoculation of pathogens. These factors compound the challenges in countries like Ethiopia that face limited healthcare infrastructure, shortage of skilled personnel, and financial insufficiency.

Given these challenges, the WHO's guidelines for using NGT in the treatment of dehydration must be implemented to save lives. However, there is currently no information on how well clinicians in Ethiopia practice NGT rehydration therapy or the barriers that hinder its effective use. Therefore, this study aimed to assess the current practices and barriers toward using NGT for managing moderate to severe dehydration in under-five children among health centers in the Gamo Zone, Ethiopia.

## Materials and methods

### Participants

The study was conducted at health centers in Gamo Zone, South Ethiopia. The Zone has 15 woredas, with 251 health posts, 52 health centers, 4 primary hospitals, 1 general hospital, and 157 private clinics, which serve a total of 1,601,085 residents. Health centers were selected purposively based on their distance from the nearest hospital. Based on these criteria, 14 health centers, which are a 30-minute drive away from the nearest hospital, were selected. Study participants were selected using a lottery method from the pediatrics and emergency units of the selected health centers until data saturation was reached. As such, a total of 30 clinicians of varying educational backgrounds participated in the study.

### Data collection

A case-based structured survey was administered orally by a trained data collector in a one-on-one setting with participants. The survey was designed to gather basic background information, including the educational preparation and professional experience.

The survey also included a clinical scenario of a child presenting with clinical signs of severe dehydration. "The child is unable to take oral fluids and attempts at placing an intravenous line have failed." In a structured format, participants were asked to classify the degree of dehydration and then state what mode of rehydration therapy they would use. If a participant chose to attempt either oral fluids or placing an IV line, they were informed that the method was unsuccessful and that the nearest hospital is more than a 30-minute drive. This process continued until the participants chose an alternative

therapy (such as NGT), or persisted with their original choice (IV, ORS, or referral) after being reminded three times. Answers were audio-recorded, transcribed, and compared to existing guidelines. The survey was provided in both English and Amharic, depending on participant preference.

Once the clinician completed the survey regarding their current practice patterns, a qualitative, semi-structured interview was conducted by the trained interviewer regarding barriers to using nasogastric tubes for rehydration therapy. An interview guide with open-ended questions was used to elicit responses regarding both attitudes toward resuscitation practice and the use of nasogastric tubes for rehydration therapy. The interviews were recorded and transcribed. Data were collected from December 4, 2023, to February 9, 2024, and written informed consent was obtained prior to data collection.

### Data analysis

Survey results were reported using descriptive statistics, including numbers, frequencies, and percentiles of answers given. Two investigators independently reviewed transcripts of the interview using constant comparative analysis to identify emerging themes. This involved reading transcripts several times with line-by-line analysis of each transcript using open coding to elucidate meanings and processes. Axial coding was performed to make connections between categories to organize themes into causal relationships. Hypotheses and concepts were then developed inductively from the data. A theoretical framework was developed to explain barriers to the use of NGT by healthcare providers for fluid resuscitation of moderate to severely dehydrated children.

### Ethical considerations

A formal ethical approval letter was issued by the Institutional Research Review Board of Arba Minch University College of Medicine and Health Science with reference number IRB/1379/2023. Data collection commenced after obtaining permission from the health centers. Participants were informed about the study, and written consent was obtained for their participation. Their right to withdraw from the study at any time during data collection was respected, and all survey and interview transcripts were de-identified.

## Results

Thirty healthcare workers from remote health centers in the Gamo Zone participated in this study. Of these, 9 were health officers, while the remaining 21 were nurses with a 10 + 3 level. The study comprised 7 female and 23 male participants, with the majority falling in the 26–30 age group.

The participants were presented with a case scenario of a severely dehydrated 2-year-old child and asked to provide a diagnosis and treatment plan. Out of these, 21 (70.0%) correctly diagnosed severe dehydration, while 9 (30.0%) diagnosed moderate dehydration. When asked about management plans, among those who identified severe dehydration, 5 (23.8%) recommended IV resuscitation, while 16 (76.2%) opted for oral rehydration using ORS. After being informed that the child was unable to take fluids orally and that securing an IV line was not possible, treatment plans changed. A total of 7 (33.3%) participants still preferred IV treatment, 10 (47.6%) opted for a NGT, and 4 (19.1%) decided to refer the child to a hospital. Those planning a referral were reminded that the nearest hospital was over a 30-minute drive away by vehicle. One clinician insisted that IV treatment was the only option, while four participants maintained that referral was necessary, and the remaining 16 opted for NGT resuscitation (Fig 1). Two nurses suggested administering gentamicin and ampicillin before referring the patient, while another nurse stated that she would attempt IV fluids if ORS failed. Additionally, she suggested breastfeeding while referring if IV treatment was ineffective.

Among those who diagnosed moderate dehydration, 3 (33.3%) selected NGT resuscitation, 2 (22.2%) preferred IV therapy, and 4 (44.4%) chose oral rehydration therapy using ORS. However, after being informed that the child was unable to drink and that IV access was difficult, all participants ultimately opted for NGT resuscitation.

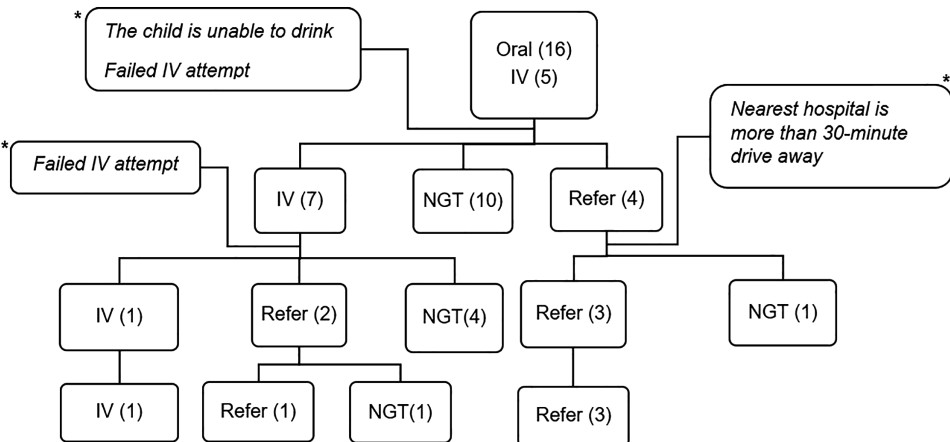

**Fig 1. Pictorial representation of treatment plans among participants who diagnosed severe dehydration, Gamo Zone, South Ethiopia.** * denote reminders given to participants until their final answer was reached.

Three participants mentioned that they would consider NGT if oral or IV therapy failed, but they noted that NGT was not commonly used at their health center. Some participants stated, *"NGT is not functional; the nurses do not use it unless doctors, health officers, or BSc nurses are available."* Others reported that *"NGTs are only available when supplied by NGOs."* Additionally, four participants did not mention that they would use NGT, although they acknowledged its potential use in dehydration management.

Clinicians generally expressed a positive attitude about NGT use, with some stating: *"NGT is an easy and effective way to treat dehydration; we just need NGT and ORS."* Others emphasized that NGT is safer and more practical than IV therapy in many cases. One clinician highlighted its life-saving role, saying, *"If we do not use NGT, children could die"*, while several participants emphasized its importance for fluid replacement in moderate to severe dehydration (Table 1). Although the participants expressed a positive attitude, they were also cautious about the complications, stating that *"insertion of NGT may injure the nasopharynx", "misplacement of the NGT into the respiratory tract"* and *"abdominal distension"* may occur.

Challenges in dehydration management were also explored. Participants identified *"late presentation"* and *"inability to drink"* as major difficulties. Other frequently mentioned barriers that contributed to late presentation was geographic inaccessibility of the health center. The participants emphasized this saying, *"the road to health center is difficult"*. Financial constraints, lack of awareness about the disease, and reliance on traditional home treatments were also mentioned as challenges to manage dehydration as these factors delay early intervention. Additionally, poor hygiene was identified as a contributing factor to the high incidence of diarrheal diseases.

**Table 1. Attitudes toward using NGT for the management of severe dehydration among healthcare professionals in health centers, Gamo zone, South Ethiopia.**

| Theme | Code (Frequency) |
| --- | --- |
| **Positive attitude** | NGT easy way for treatment (6); NGT is a good alternative to rehydrate (1); NGT is a safe route (1); NGT is important for fluid replacement(15); We are using NGT, and we will use it in the future (1); Without NGT, children would die (1) |

The most frequently raised challenge was the shortage of essential medical supplies, including resuscitation fluids, appropriately sized cannulas, and ORS. Moreover, some participants reported that even when referral is necessary, transporting children is difficult due to long distances, poor road conditions, and a shortage of ambulances. Clinician behavior was also mentioned; one participant stated that "*some clinicians disrespect and yell at parents saying why did you do this and why did you do that*" emphasizing the issue of clinicians displaying disrespect towards parents. Additionally, they stated that "*some clinicians do not want to treat the children even if they are trained*" and "*some do not check store and dispensary for the equipment carefully,*" implying their negligence (Table 2).

Additional challenges specific to NGT use were identified. Some participants reported institutional restrictions, stating that "*NGT is only permitted in hospitals.*" Others mentioned frequent shortages of NGT, noting that sometimes the "*appropriate sizes for children are not available.*" While others pointed out that available NGT sometimes expired due to underutilization. Another concern was the lack of trained clinicians, as many healthcare workers frequently leave remote health centers due to transfers or resignations. Furthermore, many expressed that there is no formal training on dehydration management using NGT and that the education received in school is insufficient for practical application. Moreover, some participants stated "*if there is ulcer or deformities in the nasopharynx we cannot insert NGT*" expressing their concern regarding contraindications. In addition, some participants feared the complications when using NGT for fluid resuscitation noting that "the *tube could be placed in the respiratory tract and the nasopharynx could be injured* ". Family resistance was also a barrier, with some parents refusing NGT insertion due to fear and lack of awareness. Parents expressed concerns such as, "*I won't let my child suffer any more; I will accept what God dictates,*" and "*The child is not going to survive.*" Some children were also uncooperative during NGT insertion and tended to remove the tube unless closely supervised (Table 3)

**Table 2. Challenges in managing severe dehydration in health centers Gamo Zone, South Ethiopia, 2024.**

| Theme | Code (Frequency) |
|---|---|
| **Health center related** | Shortage of equipment (36); Geographic inaccessibility of the HC (4); No material dispensary (2); No admission room (1); Inadequate payment (1) |
| **Socioeconomic factors** | Late presentation (9); Poor hygiene (5); Inability to afford the treatment (2); Traditional home treatment (1); Unavailability of clean water (1) |
| **Clinician related** | Clinicians negligence (3); Disrespecting patients (1) |

**Table 3. Barriers toward using nasogastric tube in managing moderate to severe dehydration in health centers Gamo Zone, South Ethiopia, 2024.**

| Theme | Code (frequency) |
|---|---|
| **Health center and clinician related** | Shortage of NGT (16); Unavailability of capable clinicians (13); No NGT training (9); Skill gap (9); Not possible in HC (4); Clinicians turn over (1) |
| **NGT Complications and contraindications** | Insertion into respiratory tract (6); It may injure the nasopharynx (5); We cannot use NGT if there is Ulcer in the nasopharynx (5); Fluid overload (4); We cannot use NGT if there is deformity in the nasopharynx (3); Abdominal distension (2) |
| **NGT drawbacks** | Less cooperation (6); Withdrawal of the tube (4); Discomfort (2) |
| **Socioeconomic factors** | Family resistance for NGT (12) |

## Discussion

The present study assessed the management of moderate to severe dehydration and the utilization of NGT in remote health centers located more than a 30-minute drive away from the nearest hospital. Our findings highlighted critical issues, such as misdiagnosis, inappropriate treatment choices, challenges that hinder effective dehydration management and various barriers towards using NGT to treat dehydration in under-five children.

A particularly concerning finding in our study was that 30.0% of clinicians misdiagnosed severe dehydration as "moderate", despite the presence of clear clinical signs and symptoms such as sunken eyeballs, prolonged skin pinch, and an inability to drink. This diagnostic inaccuracy may result from a lack of continuous training and capacity building for clinicians, as the participants themselves stated. The World Health Organization (WHO) has ascertained that misdiagnoses of emergency cases can result in delayed or inappropriate treatment, which increases the risk of complications and mortality [11]. This implies that addressing diagnostic errors should be a priority in improving treatment outcomes.

The WHO recommends IV rehydration for the management of severe dehydration in health centers, when referral to hospital is not feasible. However, when IV resuscitation is not possible and referral to a higher-level facility would take more than a 30-minute drive; NGT administration is a viable alternative [5]. Despite this, our findings showed that 76.2% of clinicians initially opted for oral treatment, which is inappropriate for children diagnosed with severe dehydration. Many clinicians adjusted their approach only after being reminded of the inability to administer fluid orally and the difficulty of accessing an IV line, with 16 choosing to use NGT. However, five clinicians refrained from using NGT due to institutional policies or misconceptions about its feasibility in health centers, implying the need for policy reform and education [12]. When comparing our findings to a study conducted in Kenya, we observed a higher level of awareness and utilization of NGT in that setting [6]. The difference could be attributed to the fact that the Kenyan study has involved hospitals that considered alternative treatment methods, such as intraosseous access and venous cut-downs. In addition, over time, awareness and acceptance of NGT for the management of severe dehydration may have increased.

In our study, nine clinicians initially diagnosed the child with moderate dehydration and planned oral rehydration therapy in the rehydration corner of the health center. However, upon recognizing that the child is unable to drink, they changed their treatment plan to NGT rehydration. Previous studies have reported that NGT is effective in managing moderate dehydration, highlighting its accessibility and efficiency as key advantages [8,10].

Challenges contributing to the ineffective management of dehydration in these remote settings were shortages of equipment such as IV fluids, ORS, NGT, and appropriate size cannulas, which created a challenge in delivering optimal care. Other studies have also reported scarcity of medical equipment and supply disruptions in various healthcare settings, implying the need for inventory management and alternative cost-effective treatment protocols during shortages [6,12,13]. Additionally, the late presentation of patients, which respondents related to traditional treatment practices, financial constraints, and geographic inaccessibility made intervention more difficult. Our finding aligns with studies that stated the lack of awareness regarding health-related issues is a significant risk factor for delayed health-seeking behavior, leading to disease complications upon presentation [6,14]. Furthermore, when severe dehydration was recognized, referral to a hospital was often delayed due to poor road conditions, transport challenges, and a lack of ambulances. This finding is consistent with reports from studies in similar settings where emergency transport remains a significant obstacle to emergency care [6,13].

Another major issue was the gap in clinical skills; many healthcare workers mentioned lack of adequate training in NGT insertion, and high staff turnover in rural areas further exacerbated this problem, pinpointing "even *if we have NGT, we do not use it unless trained health officers, BSc. Nurse or GP are not around and they do not stay long enough*". A similar finding has been reported in western Kenya, where managing severe dehydration has been associated with a skill gap to use NGT [6]. Family resistance to NGT utilization that could be driven by fear and misinformation, also limited the acceptance of this treatment option, preventing some children from receiving potentially life-saving care. Studies have mentioned that lack of awareness about NGT has contributed for such behavior [6,14].

Despite these obstacles, most clinicians in the study acknowledged the benefits of NGT in dehydration management noting that "*it is lifesaving*". Addressing these systemic challenges through targeted interventions can significantly improve dehydration management, ultimately reducing child mortality in resource-limited settings. Particularly several participants stated that improving NGT utilization through better training programs, policy adjustments, and awareness raising campaigns could play great role.

## Conclusion

Clinicians who participated in our study were aware of the importance of NGT for the treatment of moderate to severe dehydration. However, the utilization of NGT was hindered by several factors. The major challenges identified by clinicians included a gap in clinical skills, a lack of continuous training, high clinician turnover, and institutional policies limiting NGT utilization for managing dehydration. In addition to these clinician-related and institutional policy challenges, family resistance was identified as a crucial barrier to the effective use of NGT. Hence, to improve NGT functionality in dehydration management, key strategies such as training and capacity building, policy reform, workforce retention, community awareness campaigns and further research should be implemented.

## Acknowledgments

We would like to thank Arba Minch University for the opportunity, continued support, and assistance throughout this study.

## Author contributions

**Conceptualization:** Tsegazeab Ayele, Sayih Mehari Degualem, Habtamu Wondmagegn.

**Data curation:** Solomon Kassa, Sayih Mehari Degualem, Tamirayehu Abayneh.

**Formal analysis:** Tsegazeab Ayele, Muluken Birhanu, Solomon Kassa, Tamirayehu Abayneh, Maycas Gembe, Biniyam Demisse Andarge.

**Investigation:** Sayih Mehari Degualem.

**Methodology:** Tsegazeab Ayele, Solomon Kassa, Habtamu Wondmagegn, Kidus Temesgen, Habtamu Esubalew Bezie, Tamirayehu Abayneh, Biniyam Demisse Andarge.

**Project administration:** Tsegazeab Ayele, Sayih Mehari Degualem.

**Resources:** Tamirayehu Abayneh.

**Software:** Tsegazeab Ayele, Sayih Mehari Degualem, Habtamu Wondmagegn, Habtamu Esubalew Bezie.

**Supervision:** Solomon Kassa, Habtamu Esubalew Bezie.

**Validation:** Muluken Birhanu, Habtamu Wondmagegn, Kidus Temesgen, Maycas Gembe.

**Visualization:** Kidus Temesgen, Maycas Gembe.

**Writing – original draft:** Tsegazeab Ayele, Biniyam Demisse Andarge.

**Writing – review & editing:** Muluken Birhanu, Habtamu Wondmagegn, Habtamu Esubalew Bezie.

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
