## [Decision Letter · Decision Letter 0]

16 Nov 2025

Dear Dr. Meshesha,

Thank you for submitting your manuscript to PLOS ONE. After careful consideration, we feel that it has merit but does not fully meet PLOS ONE’s publication criteria as it currently stands. Therefore, we invite you to submit a revised version of the manuscript that addresses the points raised during the review process.

We look forward to receiving your revised manuscript.

Kind regards,

Vinaya Satyawan Tari, Post doctoral fellow, (M.Sc., B.Ed., Ph.D.)

Academic Editor

PLOS ONE

Journal Requirements:

3. We notice that your supplementary figures are included in the manuscript file. Please remove them and upload them with the file type 'Supporting Information'. Please ensure that each Supporting Information file has a legend listed in the manuscript after the references list.

Reviewer's Responses to Questions

**Comments to the Author**

1. Is the manuscript technically sound, and do the data support the conclusions?

Reviewer #1: Partly

2. Has the statistical analysis been performed appropriately and rigorously?

Reviewer #1: Yes

3. Have the authors made all data underlying the findings in their manuscript fully available?

Reviewer #1: Yes

4. Is the manuscript presented in an intelligible fashion and written in standard English?

Reviewer #1: No

Reviewer #1: The authors do a good job of identifying a gap in clinical practice and evaluating the current practice, attitudes, and barriers, particularly in remote areas. The manuscript needs further clarification in areas and significant editing for grammar and spelling. Please see further details in the comments below:

Abstract:

Conclusion (line 40-42) - presents new data regarding challenges to use of nasogastric tube. Conclusion should just summarize study results and next steps. The data related to challenges in the conclusion should be presented in the results.

Introduction:

Line 55 – Please write out Nasogastric tube (NGT) or any abbreviations with the abbreviation in parentheses the first time it appears in the manuscript so that it isn’t confusing to the reader. Then you can just use the abbreviation in the remainder of the manuscript.

Editing for English grammar and language would be helpful for readability. For example, Line 57-58 states “this reliance on IV therapy could lead to out off any treatment options risks children’s lives, as the inability to secure an IV line can delay life-saving interventions.” – I’m not sure what “out off” means. Is this supposed to be “cut off”?

Line 59 stating that experienced clinicians are always promoted to urban areas seems very definitive. Do you have a reference for this? If not, I would just state that commonly, experience clinicians are promoted to urban areas.

For lines 60-64, do you have any references that may support these thoughts? Or other studies that have found these same challenges? Try to provide details related to what is known and or gaps in the literature to better introduce the reason for your study.

Methods:

Data Collection, line 100-101: for alternate therapy (either nasogastric tube) is incomplete thought. Are there other alternate therapy options that were accepted?

Results:

Further description of all the themes would be helpful under attitudes and challenges, potentially in a table with number of times it was mentioned would be helpful. There are only 3 listed in the attitudes table and several in the challenges table; however, some are not well understood.

Chart with attitudes –Were there any negative attitudes?

Chart with Challenges in managing Dehydration – there are several misspelled words. Additionally, not sure what desrespection is – can you please define somewhere or consider another word.

Discussion

There continue to be several errors in grammar and spelling throughout.

Line 188 has a misspelling (sing instead of sign).

Line 213 has a misspelling (challeng instead of challenge)

**Do you want your identity to be public for this peer review?** For information about this choice, including consent withdrawal, please see our Privacy Policy

Reviewer #1: No

---

## [Author Response · Author response to Decision Letter 1]

11 Dec 2025

Dear Editor,

Thank you for your time and support. In the journal requirements section, it says that figures are included as supporting files in the manuscript. However, we have uploaded figures (not supporting files) as a separate file.

---

## [Editor Report · Decision Letter 1]

27 Jan 2026

Dear Dr. Meshesha,

Thank you for submitting your manuscript to PLOS ONE. After careful consideration, we feel that it has merit but does not fully meet PLOS ONE’s publication criteria as it currently stands. Therefore, we invite you to submit a revised version of the manuscript that addresses the points raised during the review process.

**ACADEMIC EDITOR:**

The authors discuss "Assessment of practice and barriers towards nasogastric tube rehydration of moderate and severe dehydration due to diarrheal disease in under-five children among health centers in Gamo zone, Ethiopia." The manuscript requires minor improvement in several areas. I recommend its acceptance for publication following minor revisions. The key concerns are outlined below:

Line 181: Check grammatical errors in the line.Table 2: Please check whether it is ‘Themes’ or ‘Theme.’Please always leave a space between the number and SI unit, no space before the "%", "/", and ":" signs. Please check all units in the whole text. There are inconsistencies in the expression and format of units.Lastly, the text should be reviewed for grammatical, formatting, and punctuation errors. It is advisable to seek the assistance of a native English speaker to revise and proofread the manuscript before resubmission.

We look forward to receiving your revised manuscript.

Kind regards,

Vinaya Satyawan Tari, Post doctoral fellow, (M.Sc., B.Ed., Ph.D.)

Academic Editor

PLOS One

Journal Requirements:

Additional Editor Comments:

Dear Authors,

Greetings of the day!

The authors discuss "Assessment of practice and barriers towards nasogastric tube rehydration of moderate and severe dehydration due to diarrheal disease in under-five children among health centers in Gamo zone, Ethiopia." The manuscript requires minor improvement in several areas. I recommend its acceptance for publication following minor revisions. The key concerns are outlined below:

1) Line 181: Check grammatical errors in the line.

2) Table 2: Please check whether it is ‘Themes’ or ‘Theme.’

3) Please always leave a space between the number and SI unit, no space before the "%", "/", and ":" signs. Please check all units in the whole text. There are inconsistencies in the expression and format of units.

4) Lastly, the text should be reviewed for grammatical, formatting, and punctuation errors. It is advisable to seek the assistance of a native English speaker to revise and proofread the manuscript before resubmission.

---

## [Author Response · Author response to Decision Letter 2]

30 Jan 2026

We have accepted the comments and suggestions from the editor and corrected some issues in the manuscript accordingly. Additionally, we have revised the tables and figures.

---

## [Editor Report · Decision Letter 2]

4 Feb 2026

Assessment of practices and barriers toward nasogastric tube rehydration for moderate and severe dehydration due to diarrheal disease in under-five children among health centers in Gamo Zone, Ethiopia

PONE-D-25-31921R2

Dear Dr. Meshesha,

We’re pleased to inform you that your manuscript has been judged scientifically suitable for publication and will be formally accepted for publication once it meets all outstanding technical requirements.

Kind regards,

Vinaya Satyawan Tari, Post doctoral fellow, (M.Sc., B.Ed., Ph.D.)

Academic Editor

PLOS One
---

## [Editor Report · Acceptance letter]

PONE-D-25-31921R2

PLOS One

Dear Dr. Ayele,

I'm pleased to inform you that your manuscript has been deemed suitable for publication in PLOS One. Congratulations! Your manuscript is now being handed over to our production team.

Kind regards,

on behalf of

Dr. Vinaya Satyawan Tari

Academic Editor

PLOS One